# CONSTRUCTING A 3D SCENE FROM A SINGLE IMAGE

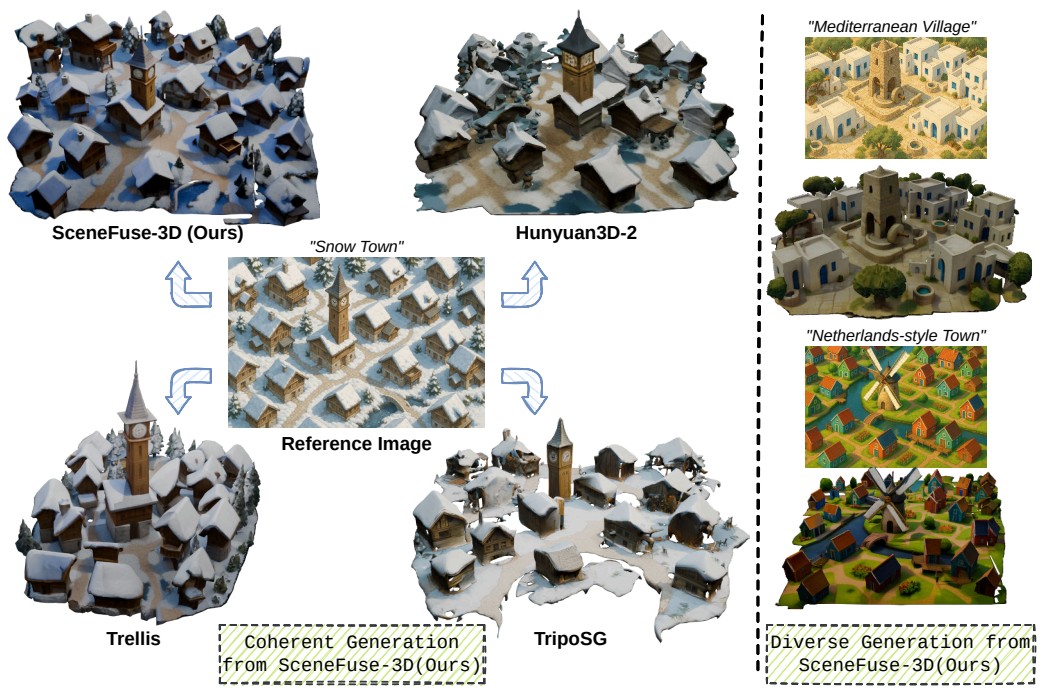

Figure 1: **3D Scene Generation from a Single Image.** Given a top-down reference image (center), SceneFuse-3D generates coherent and realistic 3D scenes that preserve geometry, texture, and layout compared to other state-of-the-art end-to-end image-to-3D generation models. Our method also generalizes across diverse styles (right), producing high-quality outputs without any 3D training.

## ABSTRACT

Acquiring detailed 3D scenes typically demands costly equipment, multi-view data, or labor-intensive modeling. Therefore, a lightweight alternative, generating complex 3D scenes from a single top-down image, plays an essential role in real-world applications. While recent 3D generative models have achieved remarkable results at the object level, their extension to full-scene generation often leads to inconsistent geometry, layout hallucinations, and low-quality meshes. In this work, we introduce **SceneFuse-3D**, a training-free framework designed to synthesize coherent 3D scenes from a single top-down view. Our method is grounded in two principles: region-based generation to improve image-to-3D alignment and resolution, and spatial-aware 3D inpainting to ensure global scene coherence and high-quality geometry generation. Specifically, we decompose the input image into overlapping regions and generate each using a pretrained 3D object generator, followed by a masked rectified flow inpainting process that fills in missing geometry while maintaining structural continuity. This modular design allows us to overcome resolution bottlenecks and preserve spatial structure without requiring 3D supervision or fine-tuning. Extensive experiments across diverse scenes show that SceneFuse-3D outperforms state-of-the-art baselines, including Trellis, Hunyuan3D-2, TripoSG, and LGM, in terms of geometry quality, spatial coherence, and texture fidelity. Our results demonstrate that high-quality coherent

3D scene-level asset generation is achievable from a single top-down image using a principled, training-free pipeline.

# 1    INTRODUCTION

Constructing 3D scene environments serves as an essential component in simulation, robotics, digital content creation, and virtual world building. They enable scalable training for autonomous agents, immersive game environments, and rapid digital twin construction. However, constructing detailed and coherent 3D scenes typically requires either expensive 3D scanning equipment, multi-view data collection, or labor-intensive modeling. In contrast, generating 3D scenes from a single top-down image offers a lightweight and accessible alternative, making it possible to bootstrap comprehensive environments from minimal input.

Despite its appeal, generating a coherent and complex 3D scene from a single image presents several fundamental challenges. First, the synthesized scene must exhibit consistent geometry across novel views, which is difficult for pure volumetric rendering methods such as Neural Radiance Fields (NeRF) (Mildenhall et al., 2021) or 3D Gaussian Splatting (3DGS) (Kerbl et al., 2023). While these techniques excel at photorealistic appearance modeling, they often suffer from geometry artifacts, especially in occluded or sparsely visible regions, leading to multiview inconsistencies and structural implausibility (Li et al., 2022; Chung et al., 2023; Zhang et al., 2024a; Yu et al., 2024b). Second, the global layout of the generated scene must remain faithful to the input image. This is particularly challenging when considering the entire scene as a single asset and utilizing image-to-3D asset generators (Xiang et al., 2024; Team, 2025; Li et al., 2025b), which often fail to preserve the spatial relationships between elements, resulting in distorted or semantically misaligned arrangements. Third, the local fidelity of individual objects should align closely with the visual evidence in the input. Due to the resolution constraints of 3D representations and the domain shift from object-level training to scene-level inference, previous image-to-3D generators are prone to producing low-quality meshes and misaligning textures.

To address these challenges, we propose **SceneFuse-3D**, a training-free framework for generating complex 3D scenes from a single top-down image, by enhancing the capability of image-to-3D object generators. Our method combines two core components: region-based generation and spatial-aware 3D inpainting. Each targets specific challenges in existing pipelines. We divide the scene into overlapping regions and synthesize each independently. This modular approach enables spatial upscaling and improves local alignment by grounding generation on localized image crops. To maintain global coherence and object continuity across regions, we estimate a coarse 3D structure from monocular depth and landmark detection, forming a spatial prior. A masked rectified flow mechanism then completes missing parts while preserving known content, enhancing structural consistency and object-level fidelity throughout the scene.

Through extensive experiments, we demonstrate that SceneFuse-3D generates realistic, diverse, and geometrically consistent 3D scenes from a single top-down image. Our method significantly outperforms strong baselines, including Trellis (Xiang et al., 2024), Hunyuan3D-2 (Team, 2025), TripoSG (Li et al., 2025b), and LGM (Tang et al., 2024b), across both human preference and GPT-based evaluations. Quantitative results show notable gains in geometry quality, layout coherence, and texture fidelity, while qualitative comparisons highlight SceneFuse-3D's ability to preserve spatial structure and fine-grained detail. These results underscore the effectiveness of our modular, training-free approach to 3D scene synthesis.

Our main contributions are summarized as follows:

- We propose **SceneFuse-3D**, a training-free framework for generating structured 3D scenes from a single top-down image, leveraging pretrained object-centric generators for zero-shot scene asset synthesis.
- We develop a modular generation strategy that combines region-wise latent synthesis with spatial-aware 3D inpainting, effectively addressing resolution bottlenecks, image-geometry misalignment, and inter-region inconsistency.
- We conduct comprehensive evaluations on diverse scenes and show that SceneFuse-3D outperforms state-of-the-art baselines in geometry quality, layout coherence, and texture realism under both human and GPT-4o-based assessments.

## 2 RELATED WORK

**3D Scene Generation with 2D Generative Models** Progress in 2D generative models (Rombach et al., 2022; Ho et al., 2020; Chang et al., 2024; Ramesh et al., 2021) has enabled pipelines that outpaint views and then reconstruct 3D via depth fusion or volumetric representations such as NeRF (Mildenhall et al., 2021) and 3DGS (Kerbl et al., 2023). Early work focused on indoor scenes (Wiles et al., 2020; Koh et al., 2021; Höllein et al., 2023; Koh et al., 2023), with later methods tackling natural scenes (Li et al., 2022; Cai et al., 2023; Fridman et al., 2023; Chung et al., 2023) and improving reconstruction through stronger depth pipelines (Yu et al., 2024b; Zhang et al., 2024c;a; Yang et al., 2024; Shriram et al., 2024; Engstler et al., 2024; Yu et al., 2024a). Despite compelling renders, these approaches often exhibit geometric inconsistency due to hallucinations, especially in occluded regions. Panoramic (Stan et al., 2023; Li et al., 2024b; Wu et al., 2023; Schult et al., 2024; Liang et al., 2024) and multiview generation (Liu et al., 2024; Tang et al., 2023c; Gao et al., 2024) improve coverage but typically restrict camera motion and still struggle with accurate geometry. Concurrently, Syncity (Engstler et al., 2025) assembles block-wise 3D generations yet produces compact layouts and does not take full scene images as input. In contrast, we directly generate geometry-consistent scene assets with arbitrary layout from a single top-down image.

**3D Scene Generative Model** Beyond 2D-to-3D pipelines, recent work directly generates scenes in native 3D. One line uses LLMs (Feng et al., 2023; Zhou et al., 2024; Çelen et al., 2024; Hu et al., 2024; Sun et al., 2023) or diffusion models (Tang et al., 2024a; Lin & Mu, 2024; Zhai et al., 2024; Vilesov et al., 2023; Maillard et al., 2024) to predict scene layouts that are then populated with assets. These methods yield semantically plausible arrangements but are often limited to predefined categories, struggle with coherent background geometry, and can suffer inter-object collisions. A separate line models scenes directly in latent 3D spaces (Wu et al., 2024; Meng et al., 2024; Ren et al., 2023; Liu et al., 2023b; Lee et al., 2024; Chai et al., 2023; Yan et al., 2024; Lee et al., 2025), e.g., BlockFusion's triplane blocks (Wu et al., 2024) and LT3SD/XCube's TUDF/voxel representations (Meng et al., 2024; Ren et al., 2023). While architecturally strong, these approaches require large, domain-specific 3D datasets (e.g., indoor rooms or urban layouts), limiting generalization to unseen scene types. In contrast, our method is *training-free* and modular, generating diverse 3D scene assets directly from a single top-down image.

**Image-to-3D Asset Generation** A parallel line of work generates *single* 3D assets from one image. Early methods combine 2D diffusion with NeRF/3DGS optimization (Poole et al., 2022; Lin et al., 2022; Tang et al., 2023b; Liu et al., 2023a; Tang et al., 2023a; 2024b) but often trade speed for geometry fidelity. Newer approaches directly generate 3D latents via diffusion (Gupta & Gupta, 2023; Xiong et al., 2024; Li et al., 2024a; Vahdat et al., 2022; Zhang et al., 2024b), and rectified-flow models further improve quality/efficiency (Xiang et al., 2024; Team, 2025; Li et al., 2025b;a). However, these methods are trained on large object-centric datasets (e.g., Objaverse-XL (Deitke et al., 2023)), so applying them to full scenes faces limited 3D resolution and domain shift between objects and scenes, leading to spatial inconsistencies and layout hallucinations. We build on pretrained rectified-flow generators (Xiang et al., 2024) and mitigate these issues with a *region-based* strategy plus *spatial-aware 3D inpainting*, yielding scene assets with high geometric fidelity and global layout coherence from a single top-down image.

## 3 METHOD

Given a single top-down image of an unknown scene, our objective is to synthesize a high-quality 3D scene that is geometrically and visually consistent with the input view. Figure 2 illustrates an overview of our pipeline.

From the scene image, we first construct the scene latents with structured latent representations (Sec. 3.1) from spatial priors (Sec. 3.2). Then, we divide the scene-level latent into region-level latents for sequential processing. For each region, we extract the latents from the latest scene-level structured latents and take those as priors for spatial-aware 3D completion (Sec. 3.3) by using the base 3D generator, Trellis (Xiang et al., 2024). Later, we fuse the updated region latents to the scene latents for cross-region consistency (Sec. 3.4). After finishing all regions, we leverage pretrained

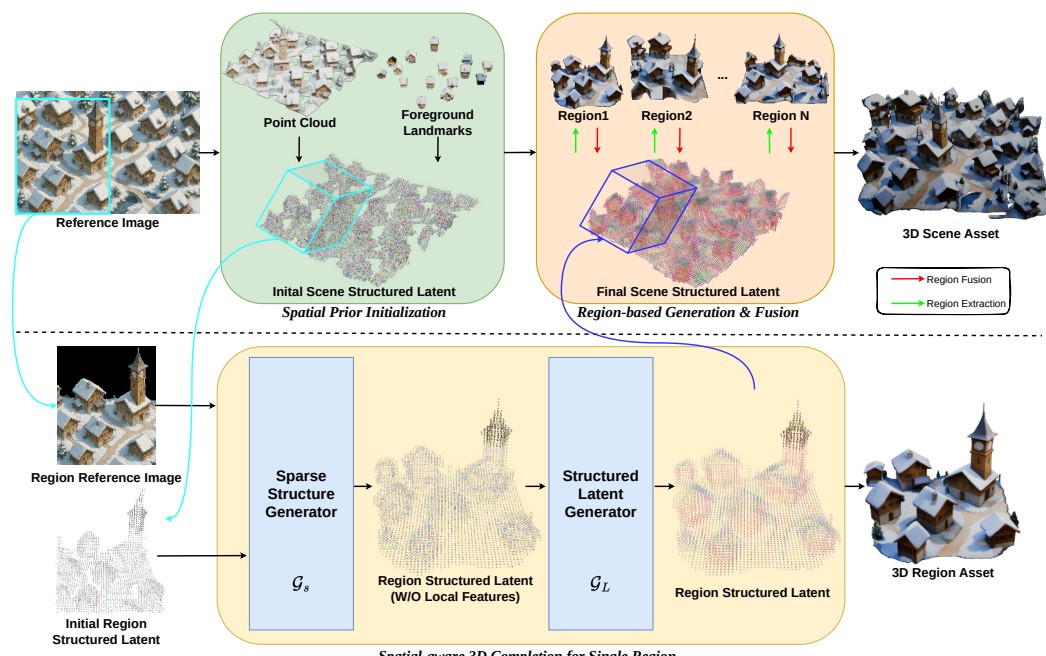

Figure 2: **Overview of the SceneFuse-3D Pipeline.** Given a single top-down image, we first estimate a coarse scene structure via monocular depth and landmark extraction to initialize the scene latent (*Spatial Prior Initialization*). The scene is divided into overlapping regions for localized synthesis and progressively fused into a coherent global latent (*Region-based Generation & Fusion*). Each region is completed using a two-stage masked rectified flow pipeline with a sparse structure generator $\mathcal{G}_s$ and a structured latent generator $\mathcal{G}_L$ (*Spatial-aware 3D Completion*). The final 3D scene is decoded from the completed structured latent.

object decoders on the complete scene structured latents to obtain a 3D scene asset. Details are explained below.

## 3.1 STRUCTURED LATENT REPRESENTATION

To leverage the pretrained knowledge of the base 3D generator, we construct the scene with a structured latent representation $z$ (Xiang et al., 2024):

$$z = \{(p_i, f_i)\}_{i=1}^L, \quad p_i \in \{0, 1, \dots, K-1\}^3, \quad f_i \in \mathbb{R}^C, \tag{1}$$

where $p_i$ denotes the positional index of an active voxel in the 3D grid, and $f_i$ represents the associated latent feature vector of dimension $C$. Here, $K$ is the resolution of the voxel grid, and $L$ is the total number of active voxels. In general, $p_i$ captures the coarse structural layout of the object, while $f_i$ encodes fine-grained local appearance and shape information. For the pretrained models, the resolution $K$ should be equal to $N = 64$. To upscale the resolution for the scene, we construct scene structured latents with resolution $M$, where $M > N$ ($M = 2N$ by default).

For the general image-to-3D asset generation, there are two pretrained rectified flow transformers: a sparse structure generator $\mathcal{G}_s$ and a structured latent generator $\mathcal{G}_L$. At inference time, $\mathcal{G}_s$ first generates the active voxel positions $\{p_i\}^L$ from the noisy grids $V_T$ with Gaussian noise. These positions are then used by $\mathcal{G}_L$ to generate the corresponding latent features $\{f_i\}^L$ from noisy features $F_T$. Both generators are conditioned on an input image condition $C_I$, encoded by DINOv2 (Oquab et al., 2023). Finally, the structured latent representation $z$ is decoded into a 3D object $O$ using object decoders, which include different sparse 3D VAE decoders for 3D Gaussians, Radiance Fields, and mesh generation. The overall process is described as follows:

$$\{p_i\}^L = \mathcal{G}_s(V_T \,|\, C_I), \quad V_T \sim \mathcal{N}(0, I)^{[N,N,N]} \tag{2}$$

$$\{f_i\}^L = \mathcal{G}_L(F_T \,|\, C_I, \{p_i\}^L), \quad F_T \sim \mathcal{N}(0, I)^{[L,C]} \tag{3}$$

$$O = \text{ObjectDecoder}(z), \quad z = \{(p_i, f_i)\}_{i=1}^L \tag{4}$$

## 3.2 SPATIAL PRIOR INITIALIZATION

Given a top-down image, the image-to-3D generator may produce outputs in arbitrary orientations. To resolve this ambiguity and provide a consistent structural prior, we initialize the scene using point clouds. Specifically, we employ a monocular depth estimator (Wang et al., 2025) to predict a depth image and infer camera parameters, from which we construct pixel-wise point clouds. Due to occlusions, these point clouds contain missing regions, which will later be filled by the image-to-3D generator.

However, occluded areas of complex objects (e.g., buildings) may have multiple plausible completions. To enforce cross-region consistency in such cases, we propose first generating landmark objects independently and then conditioning subsequent generations on their geometry. To achieve this, we use Florence2 (Xiao et al., 2024) to propose landmark bounding boxes and SAM2 (Ravi et al., 2024) to extract instance masks. Each detected landmark is then processed individually using the 3D generator to obtain instance-level meshes. These meshes are aligned with the raw point clouds using ICP (Rusinkiewicz & Levoy, 2001), replacing the original landmark regions with mesh-derived point clouds.

After normalization, we voxelize the aggregated scene point clouds at resolution $M$ to obtain the initial voxelized scene, including foreground voxels $V_0^f$, background voxels $V_0^b$, and the full scene voxel set $V_0$:

$$V_0 = \{V_0^b, V_0^f\} = \{p_i\}^L, \quad p_i \in \{0, 1, \dots, M-1\}^3 \qquad (5)$$

Finally, we initialize the corresponding voxel features $F_0$ as zeros and construct the initial structured latent representation for the scene as $z_0 = \{(V_0, F_0)\}$, shown in the top-left of Figure 2.

## 3.3 REGION-BASED GENERATION

We adopt a region-based strategy to overcome the limitations of applying pretrained object-centric 3D generators directly to full scenes. These models are trained on single-object data, where each object occupies the full latent space at a fixed resolution $N$. When extended to an entire scene, this limited capacity results in low-resolution geometry and missing details. Moreover, direct scene-level generation from a single top-down image often leads to layout distortions and semantic hallucinations, as the model struggles to maintain spatial relationships or align 3D content with image cues. To address this, we divide the scene into overlapping regions and condition each on its corresponding image crop, enabling locally grounded and high-fidelity generation.

To implement this, we divide the initial scene voxel grid $V_0$ (of resolution $M$) into a set of overlapping region-level subgrids $\{V_0^{(r)}\}_{r=1}^R$, each with shape $N^3$, where $N$ is the resolution used by the pretrained 3D generator. For each region $r$, we also extract the corresponding image crop to obtain a localized image conditioning input $C_I^{(r)}$. This ensures that the generation in each region is locally grounded in the image evidence.

**Spatial-aware 3D Completion** While region-based generation improves local fidelity, it introduces a new challenge: how to maintain global consistency, especially across overlapping regions. Furthermore, since the 3D generator may create assets from any orientation, we need to alleviate the misalignment between image conditions and region latents to enable further region fusion. To address these challenges, we draw inspiration from training-free inpainting methods in 2D diffusion models, such as RePaint (Lugmayr et al., 2022), and adapt a similar approach for 3D generation. We propose to use a masked rectified flow pipeline for 3D completion, which treats the partially completed global scene latent as a constraint and performs conditional generation over the current region by completing only the unknown parts.

Given a region-level subgrids $V_0^{(r)}$, we designate known active voxels as positions $\{p_{i,0}^{(r)}\}^{L_{r,0}}$ with corresponding latent features $\{f_{i,0}^{(r)}\}^{L_{r,0}}$. For the coarse structure generation, we define a binary mask $m_s^{(r)}$ where inactive voxels are marked for regeneration. We use the sparse structure generator $\mathcal{G}_s$ to complete the region structure and obtain active voxel positions $\{p_i^{(r)}\}^{L_r}$. Next, for the fine-grained local features generation, we retain original features for positions overlapping with known voxels; otherwise, features are initialized with Gaussian noise. A second binary mask $m_L^{(r)}$ identifies unknown

features for regeneration. We then use the structured latent generator $\mathcal{G}_L$ to obtain the inpainted local features $\{f_i^{(r)}\}^{L_r}$. Finally, we can construct the region structured latent $z^{(r)} = \{(p_i^{(r)}, f_i^{(r)})\}^{L_r}$. These two completions both leverage the masked rectified flow pipeline (Details in the next paragraph). The whole process is shown at the bottom of Figure 2 and can be formally written as follows:

$$\{p_i^{(r)}\}^{L_r} = \mathcal{G}_s(V_T \,|\, C_I^{(r)}, \{p_{i,0}^{(r)}\}^{L_r,0}, m_s^{(r)}), \quad V_T \sim \mathcal{N}(0, I)^{[N,N,N]} \tag{6}$$

$$\{f_i^{(r)}\}^{L_r} = \mathcal{G}_L(F_T \,|\, C_I^{(r)}, \{p_i^{(r)}\}^{L_r}, \{f_{i,0}^{(r)}\}^{L_r,0}, m_L^{(r)}), \quad F_T \sim \mathcal{N}(0, I)^{[L_r,C]} \tag{7}$$

$$z^{(r)} = \{(p_i^{(r)}, f_i^{(r)})\}^{L_r} \tag{8}$$

**Masked Rectified Flow for Completion**   We adopt a masked generation strategy based on rectified flow to complete the unknown regions of a structured 3D latent. Let $x_{\text{known}}$ denote the known latent values to preserve, and let $m \in \{0, 1\}$ be a binary mask that indicates which parts of the latent should be regenerated ($m = 1$) and which should remain fixed ($m = 0$).

We initialize the latent variable $x_T \sim \mathcal{N}(0, I)$ with Gaussian noise, representing the unknown region at the final time step. For each timestep $t = T, T-1, \ldots, 1$, we perform $U$ resampling steps to improve stability and smoothness (Lugmayr et al., 2022). At each resampling iteration $u$, we first compute the flow field $v_\theta(x_t, t)$ using the rectified flow model and apply an Euler update to obtain the intermediate latent:

$$x_{t_{\text{prev}}} = x_t - \Delta t \cdot v_\theta(x_t, t), \quad \text{where } \Delta t = 1. \tag{9}$$

We then re-noise the known region using a forward noise operator:

$$\texttt{forward\_step}(x, t) = (1 - t) \cdot x + [\sigma_{\min} + (1 - \sigma_{\min}) \cdot t] \cdot \epsilon, \quad \epsilon \sim \mathcal{N}(0, I), \tag{10}$$

and merge it back into the latent using the mask $m$:

$$x_{t_{\text{prev}}} \leftarrow m \odot x_{t_{\text{prev}}} + (1 - m) \odot \texttt{forward\_step}(x_{\text{known}}, t_{\text{prev}}). \tag{11}$$

$\sigma_{\min}$ denotes the minimum noise scale used by the pretrained rectified model.

If $t > 1$ and the latent needs to be resampled, we apply additional forward noise to the merged latent: $x_t \leftarrow \texttt{forward\_step}(x_{t_{\text{prev}}}, \Delta t)$.

Otherwise, we simply continue with $x_t \leftarrow x_{t_{\text{prev}}}$. This masked rectified flow process iterates until $t = 0$, at which point the completed latent $x_0$ is returned. The full procedure is outlined in Algorithm 1 in Appendix B.

### 3.4   REGION FUSION

For each generated region, we update the scene-level structured latent $z_0$ by replacing the corresponding part with the region-level latent $z^{(r)}$. Because regions are extracted using a patchification strategy, some may contain only partial observations of foreground landmarks. To preserve landmark integrity, we discard those structured latents corresponding to partial foregrounds during fusion.

Each region is extracted from the latest version of the scene-level latent, ensuring consistency across regions. If a region overlaps with previously generated ones, its overlapping voxels are constrained to match the existing content during generation. This enforces continuity and avoids inconsistencies in overlapping areas, leading to smooth transitions between adjacent regions while preserving already synthesized content.

Once all regions have been processed, the final scene-level latent is decoded using object decoders to produce scene-level meshes and 3D Gaussians. The complete textured scene is then rendered using a combination of physically based rendering (PBR) baking and Gaussian Splatting. Additional implementation details, including the patchification strategy, are provided in Appendix B.

## 4   EXPERIMENTS

### 4.1   EXPERIMENTAL SETUP

**Benchmark**   To the best of our knowledge, there is no established benchmark for 3D outdoor scene mesh generation from single images. Therefore, we construct a custom test set by prompting

Table 1: **Quantitative comparisons of SceneFuse-3D and baselines.** We report human preference win rates and GPT-4o-based weighted win rates (%) across geometry, layout, and texture quality. SceneFuse-3D consistently outperforms all baselines by large margins in both evaluations.

| | Human Preference Win Rate (Percentage) | | | GPT-4o-based Weighted Win Rate (Percentage) | | |
|---|---|---|---|---|---|---|
| Models | Geometry Quality | Layout Coherence | Texture Coherence | Geometry Quality | Layout Coherence | Texture Coherence |
| Trellis (Xiang et al., 2024) | 31.50% | 30.00% | 33.00% | 17.58% | 19.04% | 14.75% |
| SceneFuse-3D (Ours) | **68.50%** | **70.00%** | **67.00%** | **82.42%** | **80.96%** | **85.25%** |
| Hunyuan3D-2 (Team, 2025) | 33.50% | 33.50% | 31.50% | 11.66% | 12.13% | 7.67% |
| SceneFuse-3D (Ours) | **66.50%** | **66.50%** | **88.34%** | **88.34%** | **87.87%** | **92.33%** |
| TripoSG (Li et al., 2025b) | 22.50% | 23.00% | 26.00% | 24.60% | 22.34% | 22.79% |
| SceneFuse-3D (Ours) | **77.50%** | **77.00%** | **74.00%** | **75.40%** | **77.66%** | **77.21%** |
| LGM (Tang et al., 2024b) | 0.00% | 0.00% | 0.00% | 0.60% | 0.00% | 0.00% |
| SceneFuse-3D (Ours) | **100.00%** | **100.00%** | **100.00%** | **100.00%** | **100.00%** | **100.00%** |

GPT-4o (Hurst et al., 2024) to generate 100 diverse top-down scene images in a variety of styles, such as "snow village," "desert town," and more. The generation details can be found in Appendix C.

**Metrics** Without ground-truth meshes, we evaluate pairwise model comparisons per reference image on three criteria, *geometry quality*, *layout coherence*, and *texture coherence*, assessing detail fidelity, spatial arrangement, and texture–image alignment, respectively. Each pair is annotated by two AMT workers to yield a **human-preference win rate**. We also report a **GPT-4o weighted win rate**: given the same pair, GPT-4o outputs token probabilities for "A/B"; we use the probability of the chosen option as a soft vote and average over pairs. All scenes are rendered to RGB images with identical Blender settings; further GPT prompt details appear in Appendix C. Additionally, we report *rendered-view* image metrics between each method's render and its reference image: **CLIP** similarity (openai/clip-vit-base-patch32, higher is better) and **FID** (Inception-V3, lower is better), averaged over the test set; these complement the human/GPT criteria by measuring image-level fidelity.

**Baselines** We compare SceneFuse-3D with four image-to-3D generation methods: *Trellis* (Xiang et al., 2024), *Hunyuan3D-2* (Team, 2025), *TripoSG* (Li et al., 2025b), and *LGM* (Tang et al., 2024b). While *Trellis*, *Hunyuan3D-2* and *TripoSG* represent the state-of-the-art end-to-end 3D transformer models, *LGM* represents the multi-view generation-based 3D generator. All models are evaluated in a zero-shot setting using official pretrained checkpoints. To ensure fair comparison, background removal is disabled, and the full input image is encoded for all methods. We also experimented with the progressive novel-view method *WonderWorld* (Yu et al., 2025). As it does not yield a consistent editable mesh, it is not included in our mesh-based quantitative tables; we provide qualitative comparisons and discussion in Appendix A.2.

## 4.2 MAIN RESULTS

Table 1, 2 and Figure 3 present the main quantitative and qualitative comparisons between SceneFuse-3D and baseline methods. The results clearly demonstrate that SceneFuse-3D consistently outperforms Trellis (Xiang et al., 2024), Hunyuan3D-2 (Team, 2025), TripoSG (Li et al., 2025b), and LGM (Tang et al., 2024b) across geometry, layout, and texture quality, as evaluated by both human annotators and GPT-4o. These improvements can be attributed to our region-based design and spatially guided generation strategy, which together promote better alignment between image features and 3D content, while preserving scene-wide consistency.

**Quantitative Analysis** SceneFuse-3D 's region-wise decomposition aligns each latent block with a localized image crop, reducing the domain gap between object-centric training and scene-level inference. This design boosts texture fidelity, with GPT-4o assigning a 92.3% win rate versus only 7.7% for Hunyuan3D-2. The upscaled resolution further enhances structural detail, reflected in geometry improvements of +37 points over Trellis (68.5% vs. 31.5%) and +55 points over TripoSG (77.5% vs. 22.5%). Spatial priors and masked 3D inpainting stabilize layouts and smooth inter-region transitions, yielding higher layout coherence (70.0% vs. 30.0% for Trellis in human study; 87.9% vs. 12.1% for Hunyuan3D-2 in GPT-4o evaluation). LGM, leveraging multi-view image generation, fails to provide consistent geometry in this setting and collapses to oversimplified outputs.

Rendered-view metrics in Table 2 support these findings. SceneFuse-3D achieves the highest CLIP similarity (0.8030), indicating stronger semantic and visual alignment with the input than all baselines.

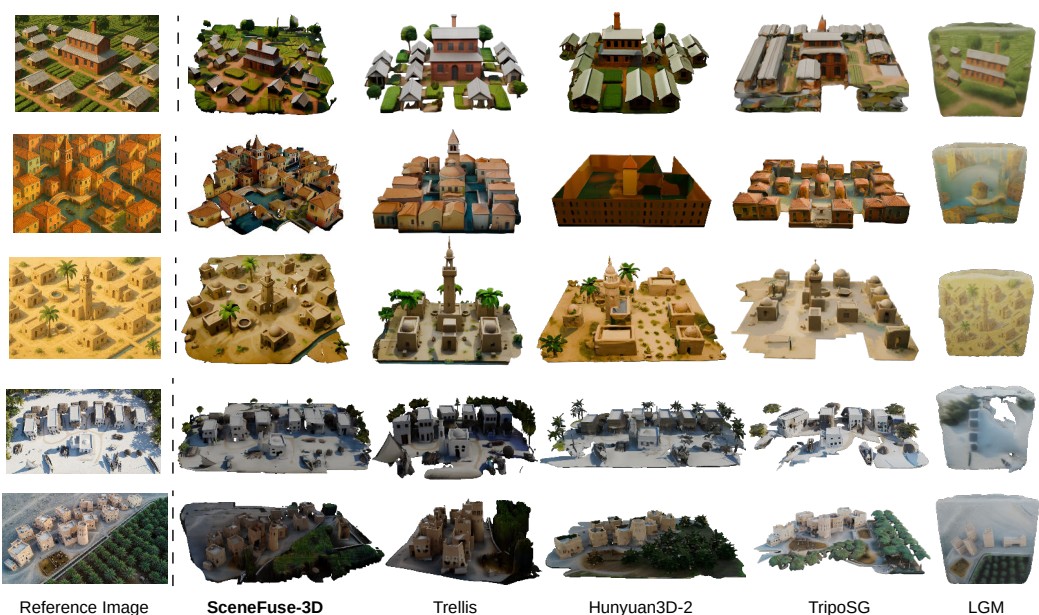

| Reference Image | **SceneFuse-3D** | Trellis | Hunyuan3D-2 | TripoSG | LGM |

Figure 3: **Qualitative comparisons between SceneFuse-3D and baselines.** Given a single top-down image (left column), we compare 3D scene outputs generated by SceneFuse-3D, Trellis (Xiang et al., 2024), Hunyuan3D-2 (Team, 2025), TripoSG (Li et al., 2025b), and LGM (Tang et al., 2024b). SceneFuse-3D consistently produces globally coherent scenes with fine-grained geometry, accurate object layouts, and realistic textures across a variety of styles and environments. In contrast, Trellis often produces oversimplified geometry; Hunyuan3D-2 suffers from structural inconsistencies and domain mismatch; TripoSG exhibits repetition artifacts and layout drift; LGM cannot generate consistent multi-view scene images for 3D construction.

Table 2: **Rendered-view metrics on the reference images.** CLIP similarity and FID are computed between each method's render and the corresponding reference image.

| Models | CLIP ↑ | FID ↓ |
|---|---|---|
| SceneFuse-3D (Ours) | **0.8030** | **258.74** |
| Trellis (Xiang et al., 2024) | 0.7760 | 288.23 |
| Hunyuan3D-2 (Team, 2025) | 0.7716 | 302.26 |
| TripoSG (Li et al., 2025b) | 0.7203 | 353.78 |
| LGM (Tang et al., 2024b) | 0.7510 | 353.86 |

It also records the best FID (258.74), showing the closest distributional match in image structure, while Trellis follows at 288.23, and the others exceed 300. Together, these results confirm that SceneFuse-3D not only excels in human and GPT-4o preference studies but also in reference-based image metrics, underscoring its advantages in both semantic fidelity and distributional realism.

**Qualitative Analysis**    Qualitatively, SceneFuse-3D produces scene assets with clear structure, consistent layout, and fine-grained surface details that closely match the reference top-down image. In contrast, Trellis often generates overly centralized, low-resolution structures and lacks peripheral detail. Hunyuan3D-2 exhibits notable issues with layout distortion and geometry hallucinations despite acceptable textures in isolated parts. TripoSG maintains some compositional structure but frequently introduces repeated objects and ignores the layout evidence within the reference image. LGM can only produce a hollow cube with the scene texture mapped onto its faces. SceneFuse-3D 's region-wise generation and spatial inpainting pipeline helps it avoid these artifacts while maintaining both global coherence and local fidelity.

These findings confirm that spatial decomposition and prior-guided inpainting are effective principles for lifting single-view image inputs into coherent, high-quality 3D scenes. Additional qualitative comparisons are available in Figure 5 in Appendix A.1.

Table 3: **Ablation Study Results.** Win rates for geometry, layout, and texture show that removing region-based generation or landmark conditioning degrades performance, highlighting the importance of both components in SceneFuse-3D.

| | Human Preference Win Rate (Percentage) | | | GPT-4o-based Weighted Win Rate (Percentage) | | |
|---|---|---|---|---|---|---|
| Models | Geometry Quality | Layout Coherence | Texture Coherence | Geometry Quality | Layout Coherence | Texture Coherence |
| SceneFuse-3D (w/o regions) | 20.00% | 17.00% | 13.50% | 6.11% | 8.48% | 6.08% |
| SceneFuse-3D | **80.00%** | **83.00%** | **86.50%** | **92.89%** | **91.52%** | **92.92%** |
| SceneFuse-3D (w/o landmarks) | 41.50% | 46.00% | 42.00% | 36.90% | 44.21% | 38.26% |
| SceneFuse-3D | **59.50%** | **54.00%** | **58.00%** | **64.10%** | **56.79%** | **61.74%** |

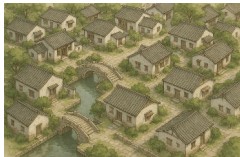 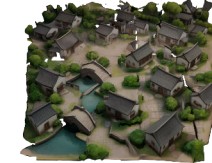 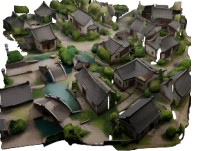 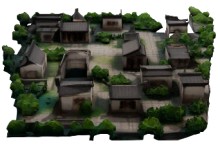

| Reference Image | **SceneFuse-3D** | SceneFuse-3D (w/o Landmarks) | SceneFuse-3D (w/o Regions) |

Figure 4: **Qualitative Ablation Results.** Left: Reference image. Middle: SceneFuse-3D without landmark conditioning. Right: SceneFuse-3D without region-based generation. Landmark conditioning ensures consistency for foreground objects, especially for objects across regions, while region-based generation preserves overall detail and coherence.

## 4.3 ABLATION STUDY

We conduct ablation studies to evaluate the contributions of key components in SceneFuse-3D: the region-based generation strategy and the use of pre-generated landmarks. Both ablation studies still apply the spatial-aware 3D completion. Quantitative results are shown in Table 3, and qualitative comparisons are illustrated in Figure 4.

**Without Region-Based Generation**    In this setting, the entire scene latent is directly passed into the pretrained 3D generator, without being split into localized regions. This leads to severe performance drops across all metrics. The results suggest that holistic generation fails to make full use of the pretrained model's capacity, which was originally trained on single-object inputs. Without localized conditioning, the model struggles to resolve spatial context and image-to-3D correspondence, producing low-resolution and spatially incoherent outputs. As illustrated in Figure 4 (right), buildings lose structural sharpness and alignment, and the overall layout becomes underspecified.

**Without Landmark Conditioning**    Removing landmark-aware initialization (depth-only prior) degrades geometry and layout, especially around large foreground structures (e.g., gates/towers). Landmarks act as semantic and geometric anchors across patches: they fix orientation/scale and provide reliable context for masked completion. Without them, region completions drift at boundaries, yielding duplicated façades or misaligned parts across neighboring patches (Fig. 4, middle).

Overall, these ablations show complementary roles: *region-wise decomposition* keeps generation within the base model's effective receptive field and preserves local detail, while *landmark anchors* enforce cross-patch continuity and stabilize global structure—both are necessary for coherent, high-fidelity scenes.

## 5 CONCLUSION

To address the challenge of generating high-quality, coherent 3D scenes from a single image, we proposed SceneFuse-3D, a training-free framework that decomposes scenes into overlapping regions and guides generation with spatial priors. A spatial-aware 3D completion with masked rectified flow preserves local object fidelity while enforcing global coherence. Empirically, SceneFuse-3D outperforms existing methods across geometry, texture, and layout, underscoring the promise of modular, spatially grounded generation for stereotype 3D scene synthesis from minimal input.

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

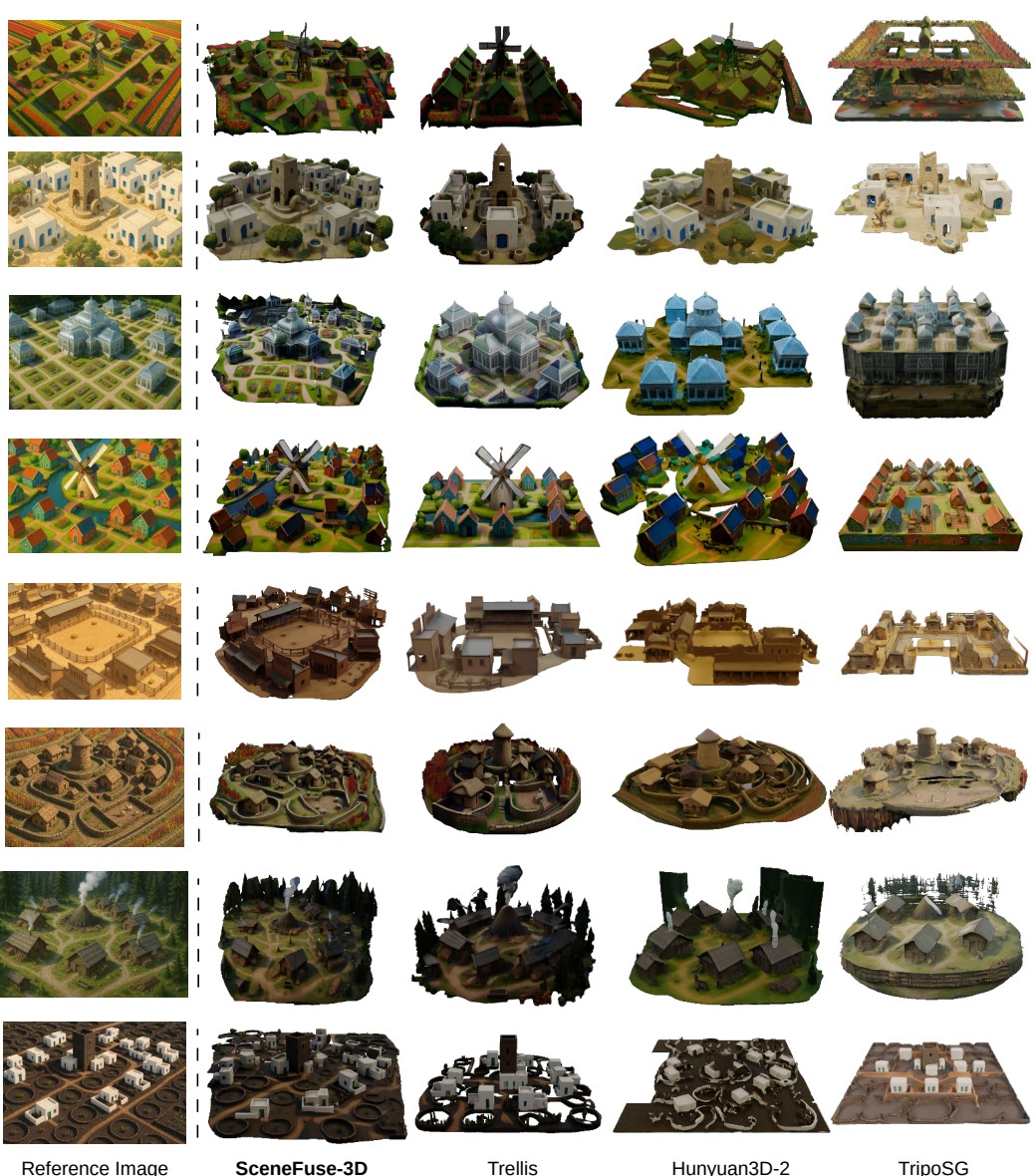

Reference Image     **SceneFuse-3D**     Trellis     Hunyuan3D-2     TripoSG

Figure 5: **More qualitative comparisons between SceneFuse-3D and baselines.** From the image, we can find that SceneFuse-3D can generate more coherent scenes from diverse scene images. LGM is skipped since it fails to generate structured scenes for all inputs.

## A MORE RESULTS

### A.1 ADDITIONAL QUALITATIVE RESULTS

Figure 5 presents additional qualitative results comparing SceneFuse-3D with Trellis (Xiang et al., 2024), Hunyuan3D-2 (Team, 2025), and TripoSG (Li et al., 2025b) across a diverse set of visual scenes. Besides, we collect some top-down view images from the internet for qualitative comparisons on real image inputs, shown in Figure 6. These examples further demonstrate the robustness and generality of our approach across different architectural styles, spatial layouts, and artistic domains.

SceneFuse-3D consistently produces scene assets that are geometrically detailed, visually coherent, and well-aligned with the input reference images. In contrast, baseline models frequently suffer from artifacts such as repeated structures, layout collapse, or low-resolution textures. These comparisons

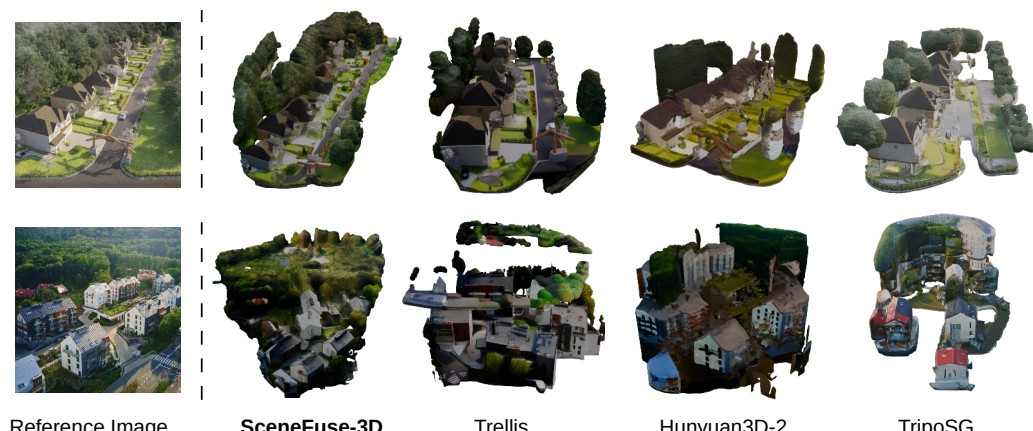

Reference Image    **SceneFuse-3D**    Trellis    Hunyuan3D-2    TripoSG

Figure 6: **Qualitative comparisons between SceneFuse-3D and baselines in real images.** We sample some top-down view images from the internet for comparisons. From the image, we can find that SceneFuse-3D can generate more coherent scenes.

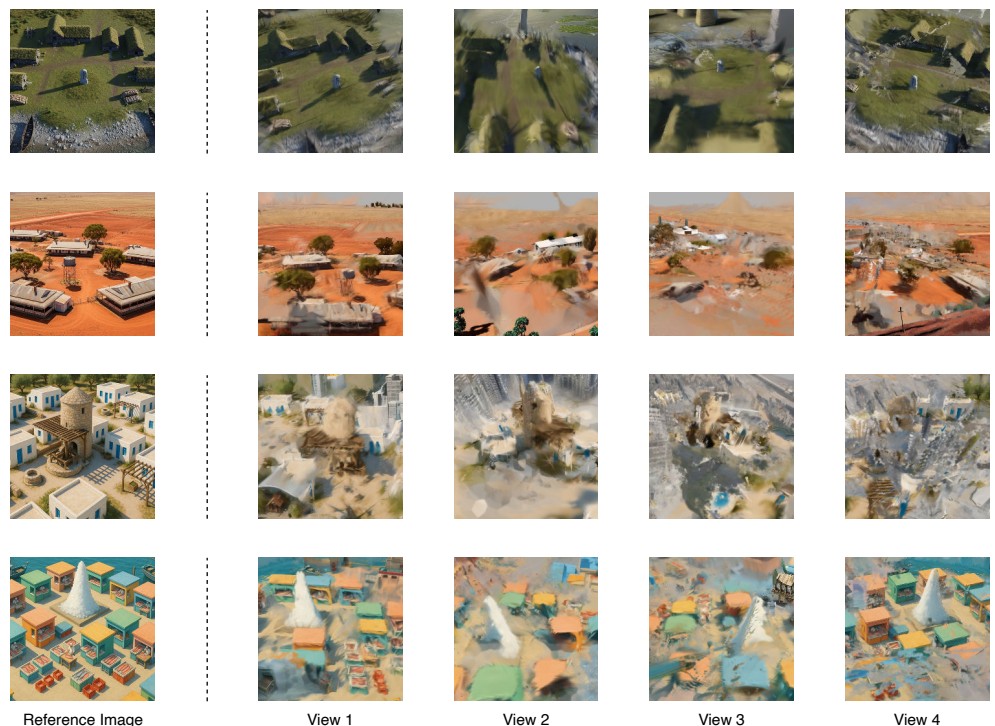

Reference Image    View 1    View 2    View 3    View 4

Figure 7: **Results of the WonderWorld.** As long as the viewpoint changes, we can find that WonderWorld cannot generate consistent geometry information for 3D scene generation.

further highlight the benefits of our region-based generation and spatial-aware inpainting pipeline in generating diverse 3D scenes from a single image without training.

---

**Algorithm 1:** Masked Rectified Flow for Completion Pipeline

---

**Input** : $v_\theta(x, t)$ — learned flow field;   $x_{known}$ — known latent to preserve;
  $m \in \{0, 1\}$ — mask for regeneration (1=regenerate, 0=preserve);
  $T$ — total steps;   $U$ — Resample times per step;   $\sigma_{\min}$ — minimum noise scale

**Output** : $x_0$ — regenerated latent

/* Forward-noise operator                                    */
$\epsilon \sim \mathcal{N}(0, I)$
$\text{forward\_step}(x, t) = (1 - t)\,x + \left[\sigma_{\min} + (1 - \sigma_{\min})\,t\right]\epsilon$

/* Initialization                                            */
$x_T \sim \mathcal{N}(0, I)$;
**for** $t = T, T-1, \ldots, 1$ **do**
  $t_{\text{prev}} \leftarrow t - 1$;
  $\Delta t \leftarrow t - t_{\text{prev}}$;
  **for** $u = 1, \ldots, U$ **do**
    $v \leftarrow v_\theta(x_t, t)$ ;                    /* predict flow field */
    $x_{t_{\text{prev}}} \leftarrow x_t - \Delta t\, v$ ;            /* Euler update on unknown */
    $\hat{x}_{t_{\text{prev}}} \leftarrow \text{forward\_step}(x_{\text{known}}, t_{\text{prev}})$ ;        /* re-noise known */
    $x_{t_{\text{prev}}} \leftarrow m \odot x_{t_{\text{prev}}} + (1 - m) \odot \hat{x}_{t_{\text{prev}}}$;
    **if** $u < U$ **and** $t > 1$ **then**
      $x_t \leftarrow \text{forward\_step}(x_{t_{\text{prev}}}, \Delta t)$;
    **else**
      $x_t \leftarrow x_{t_{\text{prev}}}$;

**return** $x_0 = x_0$;

---

## A.2 ADDITIONAL BASELINE

WonderWorld (Yu et al., 2025) enables novel view synthesis of 3D scenes from a single image through text-guided inpainting, which is designed to be effective when a coarse single-view Gaussian representation is initialized and can be refined and enriched across different views. In our task of constructing a consistent 3D scene, we set up the camera trajectory to rotate along the vertical normal to the horizontal plane of the reference images for a comprehensive scene synthesis. However, WonderWorld struggles to generate view-consistent structures and often fails to update the Gaussians effectively due to suboptimal inpainting results based on the existing observations, shown in Fig. 7.

## B METHOD DETAILS

**Region Extraction Strategy.**   To generate overlapping regions with balanced seams, we first compute the tight bounding box of all occupied voxels and tile it with a base grid of non-overlapping patches of size $(p_x, p_y, p_z)$. For the vertical ($z$) axis, start positions are evenly interpolated so that the entire height is covered with the minimum number of full patches. Next, we insert *seam patches* to equalise overlap: for every pair of neighbouring base-grid origins we create an additional patch whose origin is the midpoint between them—first along the $x$-axis (keeping $y, z$ fixed), then along the $y$-axis (keeping $x, z$ fixed). The union of base and seam origins is deduplicated and sorted, after which the corresponding voxel sub-volumes are extracted. The procedure returns the list of patch origins and their binary masks, and is invoked once per scene to define the region schedule used in our region-wise generation and fusion pipeline.

**Masked Rectified Flow Algorithm**   For completeness, we provide the full algorithmic details of the masked rectified flow completion process used in our spatial-aware 3D inpainting pipeline. While the core formulation is introduced in Section 3.3, this pseudocode (Algorithm 1) clarifies the iterative update, re-noising, and resampling procedures that enable conditional generation of unknown regions while preserving the known latent structure.

Table 4: Summary of the three-stage SceneFuse-3D pipeline.

| Stage | Main function | Components / status |
|---|---|---|
| Scene Latent Initialization | Build spatial priors and a scene-level structured latent from a single top-down image | Monocular depth, Florence2, SAM2, point-cloud alignment and voxelization into a Trellis-style structured latent; mostly off-the-shelf with a scene-level extension (ours). |
| Region Generation | Region-wise sparse structure and latent feature completion in 3D | Trellis $G_s$ and $G_L$ applied on cropped regions with masked rectified flow and local image conditioning; our region-based masked 3D completion built on Trellis. |
| Region Fusion | Enforce cross-region consistency and decode the final 3D scene | Landmark-aware fusion of the scene latent (ours) plus Trellis object decoders and standard rendering (off-the-shelf). |

**Implementation Details**   For all rectified flow generators, we step the sampling time step $T = 50$. The classifier-free guidance scales are 7.5 and 5 for the spare structure generator and structured latent generator. Resampling time $U$ during the masked rectified flow is set to 2. The whole pipeline can be loaded with an NVIDIA RTX A5000 GPU with 24G VRAM.

**Runtime Analysis**   All timing measurements are obtained on NVIDIA RTX A5000 GPUs with 24G VRAM, using the same software stack as in our main experiments. The total runtime of SceneFuse-3D depends primarily on the number of detected landmarks and the number of non-empty regions. In our current implementation, we select at most 10 landmark instances; each landmark requires running a full Trellis image-to-3D generation followed by point-cloud alignment, which takes roughly 1 minute per landmark. For the scene body, we divide the scene into 8 overlapping regions and apply the sparse structure generator and structured latent generator with masked rectified flow to each region, resulting in approximately 40 seconds per regional completion. In practice, depending on how many landmarks are actually detected and how many regions contain active voxels, the end-to-end runtime per scene ranges from about 4 minutes (few landmarks, sparse regions) to about 14 minutes (many landmarks, dense regions). For comparison, under the same hardware and resolution settings, a single-pass run of Trellis takes about 50 seconds, Hunyuan3D-2 about 2 minutes, and TripoSG about 1.5 minutes for one scene. While our region-wise pipeline incurs additional overhead compared to these one-shot generators, it enables significantly improved geometry quality and layout coherence at scene scale by reusing a fixed-capacity object-centric backbone.

**Pipeline Decomposition and Modularity**   To clarify which components of our method are reused from existing work and which are newly introduced, we decompose SceneFuse-3D into three stages: *Scene Latent Initialization*, *Region Generation*, and *Region Fusion*. In the first stage, we build spatial priors and a scene-level structured latent from a single top-down image using off-the-shelf monocular depth estimation, Florence2, and SAM2, followed by point-cloud alignment and voxelization into a Trellis-style structured latent representation. In the second stage, we perform region-wise sparse structure and latent feature completion in 3D by reusing Trellis's sparse structure generator $G_s$ and structured latent generator $G_L$, but adapting them with a masked rectified-flow scheme and local image conditioning to support spatially constrained, training-free completion at scene scale. In the final stage, we fuse all updated regions back into the global scene latent with landmark-aware fusion rules that enforce cross-region consistency and preserve key structures, and then decode the resulting latent into meshes and 3D Gaussians using Trellis object decoders and standard rendering. Table 4 summarizes these three stages and highlights which parts are off-the-shelf and which are introduced in this work.

## C  EXPERIMENT SETTINGS

**Test Set Generation**  To evaluate models' performance under stylistically diverse conditions, we curated a human-verified synthesized image test set with 100 top-down views. Given an example image, we ask ChatGPT-o3 (OpenAI, 2025) to generate image prompts for top-down scene views with these requirements: *1280 × 720 resolution, quasi-orthographic three-quarter ("isometric") camera, one hero landmark at the image centre, 10–20 surrounding buildings, daylight illumination, and "no far-away object"*. Then, we used GPT-4o (Hurst et al., 2024) to generate scene images according to the corresponding prompts.

**Human Evaluation**  Given a reference image and observations of two generated scenes, we ask the human annotator to answer these three questions:

- Which scene, A or B, has geometry that is more detailed, precise, and closer to the reference image?
- Which scene, A or B, demonstrates a spatial layout and arrangement of objects that is more coherent and closely aligned with the layout in the reference image?
- Which scene, A or B, exhibits textures that are significantly more coherent and consistent with the reference image?

**GPT-4o-based Evaluation**  For GPT-based automatic evaluation, we ask the same questions as the human evaluation and prompt the model to directly return the answer with top-5 token log probability. Then, we extract the token probability of 'A' and 'B' (0 if not included) as the answer weights. We treat $P(\text{A})$ as a soft vote when our method is option A (and analogously for B). Weighted win rate is computed as $P(\text{win})$ if our model wins, and $1 - P(\text{lose})$ otherwise, then averaged across all pairs.

## D  LIMITATION

Although SceneFuse-3D delivers strong scene-level results, several challenges remain. The pretrained 3D generator we adopt is trained on single-object imagery; even after region decomposition, the underlying distribution mismatch can lead to patch-level hallucinations—for example, duplicated façades, unrealistic roof shapes, or occasionally missing geometry in otherwise well-observed areas. Future work could mitigate this via scene-level fine-tuning or domain adaptation.

Geometric holes in our outputs currently arise from two main sources. First, our coarse spatial prior contains many voids where occlusions obscure geometry; regions dominated by such unobserved space sometimes inherit empty or over-smoothed surfaces from the generator. Second, distribution shift can cause the backbone to under-fill thin or high-frequency structures even in observed regions, leading to *observed-space* holes despite valid image evidence. Integrating uncertainty-aware depth completion, multi-view cues, or semantic/structural priors may yield denser scaffolds and more reliable inpainting for both cases in future work.

