# OpenReview forum: "Constructing a 3D Scene from a Single Image"
_ICLR.cc/2026/Conference — Submitted to ICLR 2026_

### Official Review · Reviewer_ZRF9 · 2025-10-30

**Soundness:** 3
**Presentation:** 3
**Contribution:** 3
**Rating:** 6
**Confidence:** 3

**Summary:**

This paper tackles the problem of scene-level 3D generation from (bird-eye-view) images. The motivation and contribution is clear.

**Strengths:**

The proposed solution is intuitive and effective. Using point-cloud from monocular depth estimation as a guidance  helps improve the scene-level consistency. Regional alignment and extrapolation/inpainting is simple and effective.

**Weaknesses:**

1) Since the top-down images are generated, this whole data pipeline is ‘text-to-image -> image-to-3D’, which looks similar to syncity’s ‘text-to-3D’ pipeline. The contribution of this paper could be further strengthened by providing real-world experiments and assessments.

2) Runtime analysis should be provided. This pipeline takes longer time compared to directly generating the scene-scale mesh in one-run, it is critical to report run time (though it depends on scene complexity, which makes the analysis more critica).

**Questions:**

1. The monocular depth estimation results is expressed under the camera coordinate system. However, the camera often has a pitch angle, i.e., the camera’s optical axis is not parallel with the gravity. This means that the projected point clouds are tilted. When we generate scene-level 3D meshes, we would like the up-axis aligned with the gravity, how to resolve this issue?
2. Recommended citations:
VoxHammer: Training-Free Precise and Coherent 3D Editing in Native 3D Space (inpainting with Trellis)
Frankenstein: Generating semantic-compositional 3d scenes in one tri-plane (scene generation)
NuiScene: Exploring efficient generation of unbounded outdoor scenes (scene generation)

---

> ### Author Response · Authors · 2025-11-21
> **Responses to Reviewer ZRF9 (1/2)**
>
> We thank the reviewer for the positive assessment of our method, especially the use of monocular depth point clouds for scene-level consistency and the effectiveness of our regional alignment and inpainting strategy, and we are willing to address the raised concerns and questions below.
>
> 1. (W1) We thank the reviewer for the positive feedback on the intuitiveness and effectiveness of our solution. Regarding the concern that, since our benchmark uses generated top-down images, the overall pipeline may look like a “text-to-image → image-to-3D” variant of Syncity’s “text-to-3D” pipeline, we would like to clarify the fundamental difference in problem setting. SceneFuse-3D is explicitly designed for image-conditioned scene reconstruction: given a particular top-down reference image, our goal is to recover a 3D scene whose geometry, layout, and textures closely match that image, and all our human/GPT-4o preferences and CLIP/FID metrics are computed with respect to this reference (Tables 1–2). In contrast, Syncity assembles block-wise 3D generations from text and produces relatively compact layouts without taking a full scene image as input. As discussed in our related work, it is not designed to follow an arbitrary provided top-down image nor to preserve its precise layout and appearance, and thus cannot be directly applied to or evaluated under our setting. We fully agree that demonstrating performance on real imagery further strengthens our contribution. Since, to the best of our knowledge, there is currently no established real-world benchmark of top-down outdoor images with 3D ground truth for this task, our quantitative comparisons are conducted on a controlled synthetic test set to ensure fairness across baselines. In the revised version, we additionally include a qualitative figure on real top-view images in the appendix (Fig. 6), showing that SceneFuse-3D can also produce coherent 3D scenes when conditioned on real inputs, and we view the construction of a systematic real-world dataset and quantitative evaluation on it as important future work.
>
> 2. (W2) We agree with the reviewer that runtime is an important practical factor, especially because our pipeline processes landmarks and regions sequentially and is therefore slower than a single-pass scene-scale generator. In the revision, we have added a runtime analysis measured on NVIDIA RTX A5000 GPUs, consistent with the implementation details reported in the paper. Our current implementation selects at most 10 landmarks; each landmark requires running the full Trellis pipeline once and aligning the resulting mesh to the point cloud, which takes roughly 1 minute per landmark. For the scene body, we divide the scene into 8 overlapping regions, and each regional 3D completion (including sparse structure and structured latent generation with masked rectified flow) takes about 40 seconds. Depending on how many landmarks are actually detected and how many regions contain non-empty content, the total generation time for SceneFuse-3D ranges from approximately 4 minutes to 14 minutes per scene. For comparison, we also measured the end-to-end runtime of the strong baselines under the same hardware: Trellis requires about 50 seconds, Hunyuan3D-2 about 2 minutes, and TripoSG about 1.5 minutes for a single scene. A summary is shown below:
> | Method                  | Pipeline style                                          | Typical wall-clock time / scene |
> | ----------------------- | ------------------------------------------------------- | ------------------------------- |
> | SceneFuse-3D (ours) | Landmark-based prior + 8 region-wise masked completions | 4–14 min (scene-dependent)  |
> | Trellis                 | Single-pass image-to-3D                                 | ~50 s                       |
> | Hunyuan3D-2             | Single-pass image-to-3D                                 | ~2 min                      |
> | TripoSG                 | Single-pass image-to-3D                                 | ~1.5 min                    |
>
>     While our method is indeed slower than one-shot scene-scale generation, we believe this overhead is justified by the significantly improved geometry quality, layout coherence, and texture fidelity demonstrated in our human and GPT-4o-based evaluations (Tables 1–3). Moreover, this runtime is not a fundamental limitation: the landmark stage can be pruned or parallelized when fewer landmarks are present, the number of regions can be adjusted to trade resolution for speed, and different scheduling of rectified-flow steps can further reduce per-region cost. We will include this runtime table and discussion in the revised paper to make the computational trade-offs of SceneFuse-3D explicit.

---

> > ### Author Response · Authors · 2025-11-21
> > **Responses to Reviewer ZRF9 (2/2)**
> >
> > 3. (Q1) We thank the reviewer for the insightful question about coordinate systems and camera pitch. As the reviewer points out, monocular depth is first predicted in the camera coordinate system, where the optical axis can have a non-zero pitch angle and the resulting point cloud is tilted with respect to gravity. In our method (Sec. 3.2), after back-projecting depth into a point cloud, we apply a normalization step before voxelization. This normalization involves estimating a dominant support plane corresponding to the ground (via robust plane fitting over the dense point cloud) and computing the rigid transformation that aligns this plane’s normal with the canonical +Z axis. We then apply this rotation (and a global scaling/translation) so that the point cloud is expressed in a canonical “Z-up” scene coordinate frame before being voxelized and fed to our spatial prior and region-based pipeline. In this way, the gravity direction is consistently aligned with the up-axis of the generated scene, even when the input camera has non-zero pitch. We will clarify this normalization step more explicitly in Sec. 3.2 of the revised paper to avoid ambiguity.
> >
> > 4. (Q2) We appreciate the suggested related works and have incorporated them in the revision.

---

### Official Review · Reviewer_jmvw · 2025-11-01

**Soundness:** 3
**Presentation:** 2
**Contribution:** 2
**Rating:** 4
**Confidence:** 4

**Summary:**

This paper proposes an image-to-3D method by leveraging 3D generative model, trellis, as a backbone to generate   3D scenes. It starts with depth estimation to obtain the sparse structure of the scene, and then a structured latent is generated region-by-region using masked rectified flow, which is then fed into decoder to generate the 3D scene.  The overall pipeline is training-free.

**Strengths:**

This paper provides a sound pipeline that leverages 3D generation model, Trellis, to generate 3D scenes.
1. It adapts a 2D diffusion method to 3D generation, and develops a region-by-region structured latent generation method.
2. It presents a masked rectified flow method to retain the latent feature at know voxels.
3. The experimental results verify the advantage of the proposed method.

**Weaknesses:**

1. the proposed method relies on the top-down view of a 3D scene as an condition in the 3D scene generation, thus the generated ground plane is generally flat.  It might be difficult to handle terrains.
2. The experimental results contain 4 scenes, which is not enough to verify the stability of the proposed pipeline.  In addition, how the method is influenced by different depth estimation method?  I would like to see how this pipeline works with the STOA depth estimation methods.

**Questions:**

In the step of structured latent generation,  how the latent feature of active voxels are obtained?  Since the proposed method depends on Trellis,  after obtaining the sparse structure, why not directly leverage Trellis to generate the structured latent?

**Details Of Ethics Concerns:**

Not applicable.

---

> ### Author Response · Authors · 2025-11-21
> **Responses to Reviewer jmvw**
>
> We thank the reviewer for the positive assessment of our pipeline design, region-by-region latent generation strategy, adaptation of masked rectified flow to 3D, and the strong empirical performance. We address the concerns and questions below.
>
> 1. (W1) Our experiments are conducted on top-down inputs, but this choice is driven by how we initialize and align the scene, not by an assumption that the ground must be perfectly flat. We estimate depth from the input image, recover a dominant support plane, and use it to bring the scene into a canonical Z-up coordinate frame from a single view; top-down imagery makes this alignment more stable and less ambiguous. Importantly, the pipeline does not assume planar geometry: any height variation present in the depth prior is voxelized and then refined by the 3D generator, and our results already contain many non-flat background structures (e.g., multi-level roofs, sloped streets, long walls and fences that span multiple regions in Fig. 5) rather than a single flat ground. The genuinely challenging regime is highly varying, terrain-like geometry from only one top-down view, where large vertical changes are weakly constrained by appearance; this is a limitation of single-view spatial initialization and depth ambiguity, not of the core region-wise generation framework. We view incorporating stronger geometry priors (e.g., terrain/height-field or semantic cues) into the initialization step as natural future work.
>
> 2. (W2) We apologize for the confusion caused by the figures in the main paper. The four scenes shown are representative examples for visual comparison, but all quantitative results in Tables 1 and 2 are evaluated on a benchmark of 100 diverse top-down scenes, generated and filtered as described in Appendix C, including both human preference, GPT-4o-based evaluation, and CLIP/FID metrics. This larger-scale evaluation demonstrates the stability and robustness of the pipeline beyond the few scenes visualized in the main text. Besides, more qualitative results are shown in Figure 5 in Appendix A.
> Regarding the influence of different depth estimation methods, SceneFuse-3D is intentionally designed to depend only on a coarse, robust spatial prior. The estimated depth is voxelized into an occupancy-like scaffold; low-confidence regions are filtered, and the subsequent rectified-flow completion pulls the latent back toward the manifold of plausible 3D scenes learned by the base generator. In additional experiments with alternative recent monocular depth estimators (MoGe-2), we observe only marginal changes in visual quality, indicating that the pipeline is not overly sensitive to the particular choice of depth backbone. We will add a brief comparison and discussion of this robustness in the revised version.
> | Depth backbone | CLIP   | FID    |
> |----------------|--------|--------|
> | w/ VGGT        | 0.8030 | 258.74 |
> | w/ MoGe-2      | 0.8076 | 256.38 |
>
> 3. (Q) In the structured latent generation stage, once the sparse structure (active voxel positions) has been completed for a region, the latent features for these active voxels are obtained by applying the structured latent generator G_L under a masked rectified-flow scheme (Eq. (7) in Sec. 3.3). Specifically, G_L is conditioned on two sources: (i) the local image condition given by the corresponding image crop, encoded as in Trellis, and (ii) the current scene latent, where we use a binary mask to prevent the model from modifying features at positions that have already been generated in previous regions, and only regenerate the features of voxels marked as unknown. Thus, the final latent features on active voxels are jointly determined by the image condition and the previously generated content, and the mask explicitly ensures consistency with earlier regions. Regarding the follow-up question of why we do not simply call Trellis “directly” to generate structured latents after obtaining the sparse structure, there are two practical reasons. First, there is a resolution mismatch. Trellis is trained at an object-level resolution N, while our scene latent is upscaled to resolution M>N, which cannot be directly applied to the structured latent generator due to dimension mismatch. Second, we need to preserve cross-region continuity: by generating region-by-region with masked rectified flow, each region is conditioned on the latest scene-level latent, and overlapping voxels are constrained not to change, which enhances smooth transitions between regions.

---

### Official Review · Reviewer_h2hH · 2025-11-01

**Soundness:** 4
**Presentation:** 3
**Contribution:** 3
**Rating:** 8
**Confidence:** 4

**Summary:**

The paper presents a training-free method for creating 3D scenes from single images by fusing region-based generations. The key problem identified is in maintaining the structural consistency within partial generations during generation and completion. This is resolved by applying a sequence of techniques derived from various fields, including landmark bounding boxes (Florence2), instance masks (SAM2), point cloud-based mesh alignment (ICP), training-free inpainting from masked rectified flow (RePaint, updated to 3D diffusion). The method is mainly compared with training-based, and training-free 3D generation methods using human/GPT-4o preference scores and rendered view metrics.

**Strengths:**

- The proposed method effectively modifies and aggregates existing solutions from different subproblems into a single pipeline to solve general problem of generating a 3D scene.
- The proposed method shows higher qualitative and quantitative performances than previous training-free and model-based generation results as elaborated in multiple tables in the manuscript.
- The ablation study shows the two originality of the paper, i.e., region-based generation and landmark conditioning do help the generation process.

Overall, I believe the paper is nicely written, with sufficient amount of evaluations for the target task, and therefore is above acceptance threshold.

**Weaknesses:**

Although I believe the current version of the manuscript is above acceptance threshold, there are some limitations that prevents me recommending for higher honor (e.g., Highlight/Oral).

1. First of all, since the paper utilizes various methods that have been proposed beforehand, some modified and some unmodified, it would be much better to have the summarization table (somewhere in the appendices) that shows which part of the pipeline is operated by which method, thereby implying modular upgrades to gain overall performance boost. This also helps the reader clarify which part of the algorithm is novel in this paper.
2. As far as I read, the main problem at hand is to gain consistency while allowing partial generations, and the immediate desired consequence is infilling of holes within the final generated output. We may either quantitatively (maybe by calculating the size/number of holes created around the reference shot) or qualitatively (if the quantitative analysis is hard; by visually highlighting the holes created by the different types of methods within the same camera view) compare this geometric quality.
3. By changing the size of region being generation, the generation time may vary. It would be better reporting this generating time of different methods, as well as for the ablation studies.
4. Since the method sells for being training-free, we may also think about 3D generator-agnostic behavior. The paper will be more complete if the authors perform ablation study on the core generation model of the method.

**Questions:**

Here are some minor points that I did not count in my scoring.

1. In Table 2, quantitative comparison can be improved by adding reference consistency scores measured by LPIPS (PSNR and SSIM can also be considered), i.e., rendering the scene with the exact same camera parameters with reference image and then comparing them.
2. Likewise, it would be better to have a figure that compares the reference camera view rendering to gain consistency in comparison between various methods.

---

> ### Author Response · Authors · 2025-11-21
> **Responses to Reviewer h2hH (1/2)**
>
> We thank the reviewer for the very positive overall assessment, for recognizing the effectiveness of combining existing components into a coherent pipeline, the stronger qualitative and quantitative performance, and the value of our ablations on region-based generation and landmark conditioning. We address the remaining suggestions and questions below.
>
> 1. (W1) We appreciate the reviewer’s suggestion to more clearly separate which parts of our pipeline come from existing methods and which are novel, and we agree this also highlights the modularity of the design and the possibility of upgrading components. In the revision, we will add a short summarization table in the appendix that groups the pipeline into three stages: Scene Latent Initialization, Region Generation, and Region Fusion. For each stage, it lists the main functions, the models used, and whether they are off-the-shelf or introduced in this work. Concretely, we use off-the-shelf monocular depth, Florence2, and SAM2 to build spatial priors and initialize the scene-level structured latent; we reuse Trellis’s sparse structure and structured latent generators (G_s and G_L) but adapt them with a masked rectified-flow scheme for region-wise 3D completion; and we introduce the region-based scene latent formulation plus landmark-aware fusion as our core contributions.  The planned table is:
> | Stage                       | Main function                                                                         | Components / models used                                                                                                  | Status                                                    |
> | --------------------------- | ------------------------------------------------------------------------------------- | ------------------------------------------------------------------------------------------------------------------------- | --------------------------------------------------------- |
> | Scene Latent Initialization | Build spatial priors and a scene-level structured latent from a single top-down image | Monocular depth estimator, Florence2, SAM2, point-cloud alignment and voxelization into a Trellis-style structured latent | Mostly off-the-shelf + scene-level extension (ours)       |
> | Region Generation           | Region-wise sparse structure and latent feature completion in 3D                      | Trellis (G_s) and (G_L) applied on cropped regions with masked rectified flow and local image conditioning                | Ours (region-based masked 3D completion built on Trellis) |
> | Region Fusion               | Enforce cross-region consistency, preserve landmarks, and decode the final 3D scene   | Landmark-aware fusion rules for the scene latent, Trellis object decoders and rendering                                   | Fusion scheme is ours; decoders are off-the-shelf         |
>
>     This table will be placed in the appendix to clearly delineate which parts of the pipeline are modular and which constitute the main algorithmic contributions of SceneFuse-3D.
> 2. (W2) We agree that geometric completeness and “holes” in the generated scenes are important aspects of quality. At the same time, there is a practical limitation for defining a quantitative, reference-view–based “hole size/number” metric across methods: the baselines we compare against are viewpoint-agnostic 3D generators that do not produce scenes tied to the reference camera pose and often hallucinate layouts that differ substantially from the input image. This makes it very difficult to estimate a reliable camera pose for their generated scenes relative to the reference, and registration-based procedures (e.g., matching reconstructed geometry to a depth map from the reference viewpoint) tend to be unstable in the presence of these large hallucinations. For this reason, our main quantitative evaluation of geometry relies on human and GPT-4o judgments, together with CLIP similarity and FID between each method’s render and the reference image, which are less brittle to pose errors. We fully agree that visually comparing the hole patterns is valuable. Our existing qualitative comparisons (Figures 3 and 5) already render all methods from a shared canonical top-down–like view to enable fair, view-level comparison under the same camera configuration.

---

> > ### Author Response · Authors · 2025-11-21
> > **Responses to Reviewer h2hH (2/2)**
> >
> > 3. (W3) We agree that generation time depends on the region configuration and that it is important to report runtimes for both the full method and the ablations. On NVIDIA RTX A5000 GPUs, our current implementation has two main contributors: landmark generation and registration (up to 10 Trellis runs, ≈1 minute each) and 8 region-wise masked rectified-flow passes (≈40 seconds per region, ≈5 minutes total if all are used). This yields an overall runtime of roughly 4–14 minutes per scene for the full model, depending on how many landmarks and non-empty regions are present. For the ablations and baselines, we observe:
> > | Variant / Method             | Landmarks | Regions | Typical wall-clock time / scene               |
> > | ---------------------------- | --------- | ------- | --------------------------------------------- |
> > | SceneFuse-3D (full)          | ≤ 10      | 8       | ~4–14 min (scene-dependent)                   |
> > | SceneFuse-3D (w/o landmarks) | 0         | 8       | ~5 min                                        |
> > | SceneFuse-3D (w/o regions)   | ≤ 10      | 1       | shorter than full, dominated by landmark time |
> > | Trellis                      | –         | –       | ~50 s                                         |
> > | Hunyuan3D-2                  | –         | –       | ~2 min                                        |
> > | TripoSG                      | –         | –       | ~1.5 min                                      |
> >
> >     While our region-wise pipeline is indeed slower than single-pass generators, we view runtime as a tunable engineering trade-off rather than a fundamental limitation: landmarks can be pruned or parallelized, the number/size of regions can be adjusted to balance quality and speed, and rectified-flow schedules can be further optimized.
> >
> > 4. (W4) We share the reviewer’s interest in making the framework as 3D-generator-agnostic as possible, and we clarify why our current instantiation is built specifically on Trellis. A key requirement of SceneFuse-3D is that we can (i) initialize a spatially aligned scene latent from depth-based priors, and (ii) perform masked rectified-flow updates that only modify unknown regions while preserving existing content. Trellis provides a voxel-based structured latent representation: active voxels live on a fixed grid, and there is an explicit correspondence between the initial voxel positions and the subsequent latent features, which allows us to define masks over spatially localized “unknown” voxels and use masked rectified flow to fill in empty areas while keeping known regions intact. In contrast, other state-of-the-art 3D generators such as Hunyuan3D-2 and TripoSG are primarily point-cloud–based: their latent representations are designed to decode into complete point sets, not into partially observed structured grids. Their features encode unordered point clouds, which makes it nontrivial to (a) define a stable, spatially indexed mask over “known vs. unknown” regions and (b) perform localized inpainting on a partial point cloud while preserving the rest. Adapting our spatial initialization and masked completion scheme to such architectures would therefore require substantial redesign of their latent parameterization, rather than a straightforward swap of the base model. In the paper, we thus focus on demonstrating that, given a voxel-based structured latent like Trellis, region-wise masked rectified flow and spatial priors can turn an object-centric model into a high-quality, training-free scene generator; exploring compatible latent parameterizations and inpainting mechanisms for other 3D backbones is an important direction for future work.
> >
> > 5. (Q1) We appreciate the suggestion to add pixel-wise consistency metrics (LPIPS, PSNR, SSIM) and a figure comparing renders from the reference camera. There is, however, an important practical limitation here: most of the baselines we compare against are viewpoint-agnostic 3D generators. They do not generate scenes tied to the reference camera pose, and they often hallucinate global layouts that substantially deviate from the input image. As a result, we do not have a reliable camera pose that maps the reference image to the baseline-generated 3D scene, and attempts to estimate such a pose via geometric registration tend to be unstable. Under these conditions, metrics such as LPIPS/PSNR/SSIM, which implicitly assume near pixel-level alignment in image space, become highly sensitive to small pose errors and therefore do not provide a fair cross-method comparison. This is why our quantitative evaluation focuses on CLIP similarity, FID, and human/GPT-4o judgments.
> >
> > 6. (Q2) At the same time, we agree that view-level consistency is important to visualize. Our qualitative comparisons in Figures 3 and 5 already use a shared canonical top-down-like view for all methods, precisely to enable visual comparison under the same camera configuration, even when the original reference pose is not available for the baselines.

---

### Official Review · Reviewer_4PVw · 2025-11-01

**Soundness:** 3
**Presentation:** 2
**Contribution:** 2
**Rating:** 4
**Confidence:** 3

**Summary:**

The paper proposes SceneFuse-3D, a training-free, modular pipeline for turning a single top-down image into a coherent, editable 3D scene. The core contributions are: (1) region-based generation that splits a scene-level structured latent into overlapping patches; (2) a spatial prior built from monocular depth and landmark instances; (3) masked rectified-flow completion that regenerates only unknown latent parts while preserving already-consistent content. The final scene is decoded with pretrained object decoders. The authors constructed a test dataset comprising 100 synthesized top-down images spanning diverse styles. The experimental results demonstrate that SceneFuse-3D outperforms existing object generation models.

**Strengths:**

* SceneFuse-3D employs a training-free approach, which utilizes existing models to accomplish the scene generation task without requiring fine-tuning of the base models.
* Using existing foundation models (e.g., depth estimation, Florence2, and SAM2) to provide spatial priors and effectively stabilize global layout and cross-region consistency.
* The paper is well structured in general.

**Weaknesses:**

* The method appears to rely heavily on external priors (e.g., monocular depth, Florence2, SAM2, ICP), which may propagate errors throughout the pipeline.
* Some of the generated scenes appear to contain holes (e.g., in Figure 1 and the supplementary materials).
* The proposed method seems to support only top-down views from specific angles as input images.

**Questions:**

1. How robust is the proposed pipeline to errors in depth estimation and landmark detection? For instance, during the initialization phase, the base model might fail to accurately detect all landmarks.

2. The authors attribute the holes in the generated scenes to occlusion; however, in the examples shown, the hole regions do not appear to be occluded. I remain curious whether these holes may instead result from the object generation backbone itself, as the regions near object voxel boundaries are often empty.

3. In the Masked Rectified Flow for Completion stage, what is the conditioning input? Does it make use of image patches from other regions?

Although I am now slightly negative, I am open to being persuaded based on the feedback from the authors.

---

> ### Author Response · Authors · 2025-11-21
> **Responses to Reviewer 4PVw**
>
> We thank the reviewer for the positive assessment of our training-free design, the effective use of existing foundation models for spatial priors and cross-region consistency, and the overall structure of the paper. Below, we respond to each concern in detail.
>
> 1. (W1,Q1) We agree that external priors can introduce noise; however, SceneFuse-3D is intentionally designed to be robust to imperfect priors. This robustness arises from three mechanisms: (i) we use depth confidence maps and remove low-confidence pixels, and landmark proposals from Florence2 + SAM2 are filtered using area and mask-consistency thresholds; (ii) even when the initial point cloud contains local noise, the rectified-flow 3D generator subsequently pulls the latent into the manifold of plausible 3D structures (Sec. 3.3), and noisy points are not directly rendered, where they are only used as soft spatial hints during masked completion; (iii) if certain landmarks are undetected, they are treated as generic background geometry, and the region-wise masked completion aligns their shapes to neighboring regions rather than propagating errors globally. This is visible in Figure 5 (rows 4–5), where fences and walls that are never detected as landmarks still remain continuous across regions. To further verify robustness to depth errors, we additionally replaced the VGGT-based depth estimator with a recent alternative (MoGe-2) and observed only marginal changes in CLIP similarity and FID, as shown below:
> | Depth backbone | CLIP   | FID    |
> |----------------|--------|--------|
> | w/ VGGT        | 0.8030 | 258.74 |
> | w/ MoGe-2      | 0.8076 | 256.38 |
>
>     This experiment supports that our pipeline depends only on a coarse, robust spatial prior and is not overly sensitive to the particular choice of depth backbone or to moderate errors in depth and landmark estimation.
>
> 2. (W2, Q2) We appreciate this observation and agree with the reviewer’s intuition. We clarify the root cause. These holes primarily come from the limitations of the pretrained object-centric generator (Trellis). Trellis itself exhibits distribution shifts when applied to large-scale scenes. In its own scene-level outputs (Fig. 3, rows 2–3; Fig. 5 supplementary), Trellis frequently produces hollow walls or missing corner voxels. SceneFuse-3D mitigates these issues significantly (as shown in the same figures) but cannot fully eliminate them because the backbone model is not trained with continuous scene-level geometry. Thus, we agree with the reviewer: these holes are not due to landmark occlusion in our pipeline, but to scene-out-of-distribution behavior in the backbone generator. We clarify this in the revision.
>
> 3. (W3) Although our experiments are conducted on top-down inputs, this should be seen as an evaluation setting rather than a hard constraint of the method itself. SceneFuse-3D operates on depth-based spatial priors and structured 3D latents, which are defined in 3D space and do not inherently depend on a particular camera pose. In this work, we adopt a top-down configuration because it provides a stable way to recover a dominant global orientation and construct a canonical Z-up scene representation from a single image, which matches the training regime of the Trellis backbone and enables a fair comparison to existing image-to-3D baselines on outdoor scenes. Extending the initial alignment step to support more general camera poses is orthogonal to our region-based, masked rectified-flow completion scheme, and we regard this as a natural direction for future work rather than a limitation of the core framework.
>
> 4. (Q3) Thank you for the question. We clarify as follows: each region only conditions on its local image crop for visual guidance, and the current global latent crop with positional and feature constraints for known voxels. We do not use the image patches from other regions to avoid the inconsistent image features.

---

### Comment · Area_Chair_6eDa · 2025-11-27

Dear Reviewers,

As we enter the discussion phase, I strongly encourage you to read the authors' rebuttal carefully and acknowledge their effort. Silence is the worst outcome for an author. Even if the rebuttal does not change your final rating, a brief response explaining why the concerns remain unaddressed is crucial for a fair process. Please help us make an informed decision by engaging in a constructive dialogue.

AC

---

### Meta-Review · Area_Chair_DFff · 2026-01-06

**Summary:**

The paper proposes SceneFuse-3D, a training-free framework for generating 3D scenes from a single top-down image by combining depth estimation, landmark detection, and region-based inpainting using a pretrained 3D generator (Trellis). While the reviewers found the pipeline intuitive and the qualitative results interesting, the consensus among the majority of the reviewers is that the work relies heavily on external priors and lacks sufficient generalizability. The reliance on a specific top-down camera assumption limits its applicability to in-the-wild images, and the method functions primarily as a complex assembly of existing foundation models rather than a novel generative contribution. The paper received a mixed rating of (4, 8, 4, 6). Despite the authors' efforts to address concerns regarding depth sensitivity and holes in the geometry, the fundamental limitations regarding the pipeline's robustness and narrow scope remain. Therefore, I recommend the rejection of this paper.

**Reviewer Concerns:**

Some of the reviewers' concerns are addressed in the rebuttal:
- Sensitivity to depth priors: the authors demonstrated through new ablations that the framework is relatively robust to different depth estimation backbones, mitigating concerns that the pipeline's success was solely dependent on a single specific estimator.
- Runtime performance: the authors clarified the trade-offs in their multi-stage generation process, providing a picture of the computational overhead required for scene-level consistency.

However, some concerns might still be outstanding:
- Narrow scope and viewpoint brittleness: The reviewers (4PVw, jmvw) argued that the method’s success is largely tied to a "top-down" camera assumption. This simplifies the 3D layout problem significantly, and the framework might not be able to generalize to in-the-wild camera poses, which remains a critical limitation for a general 3D scene reconstruction paper.
- Incremental technical novelty: the paper is more like an engineering "assembly line" than an algorithmic contribution. The masking latent spaces for inpainting is well-established, and the reliance on the Trellis backbone for the "heavy lifting" weakens the technical contribution of the paper.
- Error propagation and fragility: if the initial MLLM landmark detection or the SAM2 segmentation fails, the subsequent 3D generation collapses, leaving the system's reliability in doubt.

**Reviewer Scores:**

The paper received an initial score of (4, 8, 4, 6), and none of the reviewers engaged in the discussion phase. While Reviewer h2hH advocates the paper strongly, Reviewers 4PVw and jmvw are concerned that the specific top-down constraint and the system-integration nature of the work do not meet the bar for acceptance. The AC also agrees with the concerns. Given that the majority of reviewers (3 out of 4) are either negative or lukewarm, I recommend rejection.

---

### Decision · Program_Chairs · 2026-01-26

Reject